# TMT-Based Proteomics Analysis of Senescent Nucleus Pulposus from Patients with Intervertebral Disc Degeneration

**DOI:** 10.3390/ijms241713236

**Published:** 2023-08-26

**Authors:** Guangzhi Zhang, Lei Li, Zhili Yang, Cangyu Zhang, Xuewen Kang

**Affiliations:** 1Department of Orthopedics, Lanzhou University Second Hospital, Lanzhou 730000, China; zhanggzh18@lzu.edu.cn (G.Z.); lil21@lzu.edu.cn (L.L.); 220220904381@lzu.edu.cn (Z.Y.); bosuye12@126.com (C.Z.); 2The Second Clinical Medical College, Lanzhou University, Lanzhou 730000, China; 3Key Laboratory of Orthopedics Disease of Gansu Province, Lanzhou University Second Hospital, Lanzhou 730030, China; 4The International Cooperation Base of Gansu Province for the Pain Research in Spinal Disorders, Lanzhou 730030, China

**Keywords:** intervertebral disc degeneration, nucleus pulposus, proteomics, tandem mass tag, bioinformatics analysis

## Abstract

Lower back pain, a leading cause of disability worldwide, is associated with intervertebral disc degeneration (IDD) in approximately 40% of cases. Although nucleus pulposus (NP) cell senescence is a major contributor to IDD, the underlying mechanisms remain unclear. We collected NP samples from IDD patients who had undergone spinal surgery. Healthy and senescent NP tissues (*n* = 3) were screened using the Pfirrmann grading system combined with immunohistochemistry, as well as hematoxylin and eosin, Safranin O, Alcian blue, and Masson staining. Differentially expressed proteins (DEPs) were identified using quantitative TMT-based proteomics technology. Bioinformatics analyses included gene ontology (GO) annotation, Kyoto Encyclopedia of Genes and Genomes (KEGG) pathway analysis, and protein–protein interaction (PPI) analyses. In addition, immunofluorescence was used to verify protein expression. In total, 301 DEPs were identified in senescent NP tissues, including 92 upregulated and 209 downregulated proteins. In GO, DEPs were primarily associated with NF-kappaB transcription factor, extracellular regions, cellular protein metabolic processes, and post-translational protein modification. The enriched KEGG pathways included TGF-β, Wnt, RAP1, interleukin-17, extracellular matrix-receptor adhesion, and PI3K/Akt signaling pathways. PPI analysis demonstrated interactions between multiple proteins. Finally, immunofluorescence verified the expressions of MMP3, LUM, TIMP1, and CDC42 in senescent NP cells. Our study provides valuable insights into the mechanisms underlying senescent NP tissues in IDD patients. DEPs provide a basis for further investigation of the effects of senescent factors on IDD.

## 1. Introduction

Intervertebral disc degeneration (IDD) is an age-related degenerative spinal disease that is the most common cause of reduced work capacity and diminished quality of life [1,2,3]. IDD can lead to various spine-related disorders, such as disc herniation, vertebral slippage, and spinal stenosis, which can cause acute or chronic lower back pain (LBP) [4]. LBP has a worldwide prevalence of 7.3%, with approximately 540 million people affected [5,6]. The intervertebral disc (IVD) is made up of a central nucleus pulposus (NP), a peripheral annulus fibrosus, and a cartilage endplate. The translucent jelly-like NP is rich in proteoglycans, type II collagen, and water. Its high elasticity and antitension properties contribute to the stability and bending functions of the spine [3,4].

Recent studies have indicated that IDD is a complex cell-mediated disease that involves NP cell senescence [4,7,8,9]. NP senescence is characterized by increased expression of p16^INK4A^ and p21^CIP^ proteins, and secretion of bioactive molecules such as MMP3, MMP13, interleukin (IL)-6, and IL-8; this is known as the senescence-associated secretory phenotype (SASP) [10,11,12,13]. SASP accelerates senescence in neighboring cells and causes intervertebral disc dysfunction and structural deterioration. In addition, various factors accelerate IDD by causing a significant reduction in the number of healthy NP cells [3,7]. Although increasing evidence suggests that inhibition of NP senescence can delay IDD progression, the underlying mechanism is unclear. Therefore, it is important to investigate its underlying mechanism to understand IDD pathogenesis and develop effective management strategies.

Proteomics studies provide important information on the molecular underpinnings of diverse biological processes. Quantitative proteomics is a high-throughput screening technique commonly used to quantify proteins and metabolites in complex samples. It can be used to identify novel targets for disease treatment and elucidate the changes in target proteins involved in disease pathogenesis and drug interventions [14,15,16,17,18,19]. Tandem mass tag (TMT)-based proteomics technology has the advantages of high throughput, high sensitivity, good reproducibility, low systematic bias, and low noise. Therefore, multiple studies have used this strategy to analyze disease-specific protein targets and understand the molecular mechanisms involved in various diseases [16,20,21,22].

We used TMT-based proteomics to investigate normal and senescent NP tissues. We screened the differentially expressed proteins (DEPs) in NP tissues of IDD patients and analyzed the underlying mechanism of action by annotating the functions and signaling pathways. This study emphasizes the molecular events involved in IDD development and provides the necessary data to develop effective therapeutic strategies.

## 2. Results

### 2.1. Screening of Senescent NP Samples

To screen for senescent NP tissues in individuals with and without IDD, we collected healthy (Pfirrmann grade I) and severely degenerated (Pfirrmann grade V) NP tissues using the MRI-based Pfirrmann grading system. Pfirrmann grade I NP tissues had a high water content and showed a high signal in T2-weighted MRI, with a more gelatinous texture seen macroscopically. Conversely, Pfirrmann grade V NP tissues showed a low signal in T2-weighted MRI and varying degrees of fibrotic changes macroscopically (Figure 1A).

In H&E staining, the tissues exhibited different characteristics with increasing IDD Pfirrmann grade. In Pfirrmann grade I NP cells, the extracellular matrix (ECM) was homogeneously dark-stained and had a few small vacuolated cells. By contrast, in Pfirrmann grade V cells, the ECM was homogeneously light-stained and had heavily aggregated and large vacuolated cells. Safranin O, Alcian blue, and Masson staining demonstrated that the ECM of Pfirrmann grade V tissues had reduced levels of proteoglycan and collagen fiber (Figure 1A). To further screen for senescent NP tissues, immunohistochemical analysis was performed to detect the expression of P16, P21, and SASP (MMP3, MMP13, IL-6, and IL-8), which indicate cellular senescence (Figure 1B,C). Western blot analysis confirmed the immunohistochemistry results (Figure 1D,E). In total, three normal and three senescent NP tissues were selected for proteomics experiments. 

### 2.2. NP Sample Quality Control

The present experiment showed that the protein concentration met the requirements of subsequent experiments (Table 1). SDS-PAGE electrophoresis was used to assess protein quality. Protein bands were consistent and free of degradation, indicating their suitability for further experimental use (Figure 2A).

### 2.3. TMT Quantitative Proteomic Analysis and Quality Control of Data

To explore the protein expression changes, TMT-based quantitative proteomics analysis was applied to characterize differentially expressed proteins (DEPs) in normal and senescent NP tissues. A total of 962 proteins were detected by TMT quantitative proteomics. We performed principal component analysis using the expression of credible proteins to determine the relationship between different samples. Each dot in the figure represents a replicate in a grouping experiment, and different colors are used to distinguish between groups. When the difference between the two samples was significant, the two coordinates were relatively distant on the score plot and vice versa (Figure 2B). Strong positive correlations were observed between the specimens within each group, whereas a negative correlation was observed between the control and experimental groups (Figure 2C). Subsequently, we calculated the Euclidean distance between samples and hierarchically clustered the sample distance matrix to obtain a hierarchical clustering dendrogram of the sample Euclidean distance (Figure 2D). In addition, we created boxplots of qualitative–quantitative analysis and density plots, as well as density plots and boxplots of relative SDs. These plots demonstrated high repeatability, with the data showing relatively small fluctuations and meeting the repeatability criteria (Figure 2E,F). 

The distribution of peptide numbers corresponds to each qualitative protein (Appendix A); the number of proteins corresponds to the molecular weights (Appendix A). During the qualitative process, we compared each peptide with a peptide in the background database; using the database search software, we obtained the coverage index of this peptide relative to the complete protein sequence. We analyzed the coverage index and rate of peptide coverage (Appendix A).

### 2.4. Identification of DEPs

DEPs between the groups were screened using fold change (FC) ≥ 1.2 and *p* < 0.05. Compared to the control group, 301 DEPs were identified in the senescent NP tissues, including 92 upregulated and 209 downregulated proteins (Figure 3A,B). Unsupervised hierarchical clustering in the R language was also performed. The cluster heatmap of the control and senescent NP tissues is shown in Figure 3C.

### 2.5. GO and KEGG Annotation Analyses of DEPs

To investigate DEP functions, we conducted GO and KEGG enrichment analyses for upregulated and downregulated DEPs. DEPs involved in biological processes were associated with NF-kappaB transcription factor, extracellular region, collagen-containing ECM, cellular protein metabolic process, and post-translational protein modification (Figure 4A–C). We used the cnetplot function in the ClusterProfiler package to visualize the DEPs enriched in cellular components, biological processes, and molecular functions (Figure 4D). In addition, we used the enrichKEGG function to identify the signaling pathways. The KEGG pathways included TGF-β, Wnt, RAP1, the regulation of actin cytoskeleton, IL-17, ECM–receptor adhesion, and the focal, hippo, and PI3K/Akt signaling pathways (Figure 5A–C). To identify the roles of DEPs in IDD, we performed KEGG pathway analysis and found that primarily the Wnt and TGF-β signaling pathways were involved (Figure 6A,B).

### 2.6. Protein–Protein Interaction (PPI) Network and Experimental Verification of DEPs

The resulting PPI network revealed direct and indirect interactions between proteins, forming a complex and extensive network of protein regulation (Figure 7A). In addition, we collected healthy (Pfirrmann grade I) and severely degenerated (Pfirrmann grade V) NP tissues using the MRI-based Pfirrmann grading system. Compared to grade I, we observed senile-like morphological changes, including broad, irregular, and slow growth, in grade V NP cells (Figure 7B). SA-β-gal staining revealed a significant increase in the senescence rate of grade V NP cells (Figure 7C). Next, we selected MMP3, LUM, TIMP1, and CDC42 for immunofluorescence verification; the expression levels of the former three were significantly increased in grade V NP cells, whereas that of CDC42 was significantly decreased (Figure 7D,E). These findings are highly consistent with the results of our proteomics analysis.

## 3. Discussion

IDD is a common age-related degenerative spinal disease. Current treatments cannot prevent its chronic progression [23,24]. Although cellular senescence can cause IDD progression, the underlying mechanism is unclear [3,7,8,9]. Proteomics studies are widely used to identify potential therapeutic targets for various diseases, including IDD [25,26]. In this study, we collected aging NP tissues from individuals with IDD for proteomics analysis and experimental validation.

In all, 301 DEPs were identified in senescent NP tissues, including 92 upregulated and 209 downregulated proteins. GO and KEGG analyses were conducted to investigate important biological processes and signaling pathways. GO analysis revealed that DEPs were significantly enriched in the extracellular region, collagen-containing ECM, cellular protein metabolic process, and post-translational protein modification. The dynamic balance between ECM anabolism and catabolism is essential to maintaining healthy NP tissues; any imbalance between these processes can lead to IDD development. Recent studies have demonstrated that levels of proteoglycan and type II collagen are significantly decreased, whereas type I collagen content is significantly increased, in IDD patients, leading to reduced water absorption, a dysfunctional mechanical microenvironment, and decreased ECM ability to withstand mechanical stress [3,4,27]. Aging is a major risk factor for IDD progression, as it leads to the accumulation of senescent cells in IVD [28]. The rate of senescence in NP cells significantly increases during IDD, and they secrete metabolic factors, such as pro-inflammatory cytokines, matrix-degrading proteases, growth factors, and chemokines, which can alter the ECM [29,30,31]. 

The enriched KEGG pathways include TGF-β, Wnt, regulation of the actin cytoskeleton, IL-17, ECM–receptor interaction, focal adhesion, and the hippo and PI3K/Akt signaling pathways [32]. Dysfunction of TGF-β family members is associated with various diseases, such as cancer, fibrosis, immune disorders, and other pathologies related to proliferation, differentiation, ECM homeostasis, apoptosis, and aging [33,34]. Notably, the TGF-β signaling pathway is important in IDD development; it inhibits ECM degradation, cell death, and inflammation, and promotes ECM synthesis and cell proliferation [35,36]. These findings suggest that this pathway protects IVD health and prevents disease. The PI3K/Akt signaling pathway is commonly involved in various biological processes, such as proliferation, senescence, and apoptosis of IVD cells [37]. Its activation delays IDD progression by increasing SOX9 expression, which promotes ECM-associated protein expression in NP cells. A previous study demonstrated that cyclic mechanical stretching ameliorates NP cell degeneration via the PI3K/AKT signaling pathway [38,39,40]. The cytoskeleton acts as a mechanotransduction mediator between ECM and NP cells, enabling the detection of changes in the NP microenvironment [41]. Cytokines can increase IL-17 production by Th17 cells in the IVD. In IDD patients, the proportion of Th17 cells and IL-17 levels are significantly increased and correlate positively with sciatic pain severity [42]. Therefore, the PI3K/AKT pathway is an important signaling pathway in IL-17-mediated IDD.

The PPI network revealed direct and indirect interactions between proteins, forming a complex and extensive network of protein regulation. We analyzed closely interacting proteins, including MMP3, PLG, ACTA1, LUM, TIMP1, HEL-S-72P, HEL113, CDC42, PPIA, and CAT. In addition, we performed immunofluorescence validation for MMP3, LUM, TIMP1, and CDC42. MMP3, a member of the matrix metalloproteinase family, is a zinc-dependent protease that can digest various substrates, including proteoglycans, gelatin, collagen, and pro-matrix metalloproteinases [43,44]. Aripaka et al. [45] demonstrated a strong correlation between histological degeneration grade and MMP3 expression in symptomatic lumbar IVDs. Lumican (LUM) is an important ECM glycoprotein in human cartilage tissue and is highly expressed in NP tissues extracted from patients with lumbar disc herniation [46]. Hayes et al. [47] demonstrated altered LUM expression in IDD. Moreover, LUM knockdown effectively alleviated TNF-α-induced inflammatory responses, cell cycle arrest, and cellular senescence.

TIMP, an anti-catabolic protein, prevents matrix metalloproteinases from breaking down proteoglycans in the IVD. TIMP1 inhibits the enzymatic activity of matrix metalloproteinase-3 [48,49,50]. Cdc42 is a small GTP-binding protein that belongs to the Rho family and is involved in the regulation of various cellular processes, such as actin cytoskeleton organization, cell polarity, cell morphology and migration, endocytosis, exocytosis, cell cycle regulation, and cell proliferation [51]. Although there is conflicting evidence regarding the role of Cdc42 dysregulation in cell and tissue aging, multiple studies have suggested that Cdc42 contributes to the development and progression of age-related pathologies, such as neurodegenerative and cardiovascular diseases, type 2 diabetes, and age-related joint and skeletal diseases [52].

This study was the first to characterize senescent NP tissues in IDD using proteomics. However, the study had some limitations. First, the results of the bioinformatics analysis were based on prespecified criteria, and changing the cut-off criteria could affect the final results. Second, we assessed changes in proteomics profiles at the protein level, not at the gene expression level. Future studies should investigate the role of DEPs in IDD development.

## 4. Materials and Methods

### 4.1. Sample Collection 

Six NP tissue specimens were collected from patients who underwent spinal surgery at the Department of Orthopedics, Lanzhou University Second Hospital. The degenerative grade of NP tissues was determined using the magnetic resonance imaging (MRI)-based Pfirrmann grading system; the NP tissues were collected during spinal surgery [53]. Normal NP tissues (grade I) were obtained from patients who underwent surgery for idiopathic scoliosis or lumbar isthmic fracture, whereas degenerated NP tissues (grades IV–V) were obtained from patients who underwent discectomy or spinal fusion. Patients with a history of tumors, tuberculosis, or infectious diseases were excluded. Informed consent was obtained from patients preoperatively.

### 4.2. Histological Evaluation 

H&E staining (G1120, Solaibao, Beijing, China), Safranin O staining (S8020, Solaibao, Beijing, China), Alcian blue staining (G1560, Solaibao, Beijing, China), Masson staining (G1340, Solaibao, Beijing, China), and immunohistochemistry (Maixin Biotech, China) was performed according to manufacturer’s instructions, and the results were recorded under a microscope (Olympus, Tokyo, Japan).

### 4.3. Western Blot

Protein samples were separated via sodium dodecyl sulfate-polyacrylamide gel electrophoresis (SDS-PAGE) and transferred to polyvinylidene fluoride membranes (Millipore Co., Bedford, MA, USA). The membranes were blocked using 5% skim milk powder and incubated overnight at 4 °C with the primary antibody. On the following day, the membranes were incubated in a secondary antibody. Protein band density was quantified using ImageJ software (verson 1.48) and the results are presented as relative levels of β-actin.

### 4.4. Protein Extraction, Digestion, and Quality Control

Protein extraction: The NP sample was mixed with pre-chilled 0.1 M ammonium acetate–methanol solution at a ratio of 5:1 and precipitated overnight at −20 °C. The precipitate was collected by centrifugation at 4 °C and 12,000× *g* for 10 min. Subsequently, it was washed twice with pre-chilled methanol at a ratio of 5:1, mixed gently, and centrifuged at 12,000× *g* to collect the resulting precipitate; the residual methanol was removed with acetone. After dissolving the dried precipitate in the sample lysate for 3 h at room temperature, the solution was centrifuged twice at 12,000× *g* for 10 min at room temperature to obtain the supernatant. The supernatant contained the total protein content of the sample, which was quantified using a BCA protein assay kit (Thermo Fisher Scientific, Waltham, MA, USA) and stored at −80 °C. 

Protein digestion: The protein solution was mixed gently with 5 mM dithiothreitol, followed by incubation at 55 °C for 30 min. It was mixed again with 10 mM iodoacetamide and incubated for 15 min at room temperature, then again with acetone at a ratio of 1:6 to precipitate the protein. It was incubated at −20 °C for >4 h or overnight. The resulting precipitate was collected by centrifugation at 8000× *g* for 10 min at 4 °C; the acetone was evaporated for 2–3 min. To re-solubilize the precipitate, 100 µL 200 Mm tetraethylammonium bromide was added. Next, a 1/50 sample mass of 1 mg/mL trypsin-TPCK was added, and the solution was made it possible to digest overnight at 37 °C.

SDS-PAGE: First, 10 μg protein was extracted from each sample and separated by 12% SDS-PAGE. The separated gels were stained with Thomas Brilliant Blue using an eStain LG Protein Stainer. Finally, the stained gels were photographed using a fully automated digital gel image analysis system.

### 4.5. Tandem Mass Tag (TMT) Labeling

TMT labeling was performed for mass spectroscopy (MS) using a TMTsixplex™ Label Reagent Set according to the manufacturer’s protocol for the 6-plex TMT kit (Thermo Fisher Scientific, USA). Briefly, 50 µL 100 mM tetraethylammonium bromide buffer was added to the myeloid histone samples and vortexed thoroughly; the labeling reaction was performed in a 1.5 mL Eppendorf tube. Subsequently, 88 µL anhydrous acetonitrile was added to the sample, vortexed for 5 min, and centrifuged. After centrifugation, 41 µL TMT reagent was added to 100 μg of the digested NP samples. The peptides were labeled with TMT-126, TMT-127, and TMT-128, respectively, for the normal NP groups, and TMT-129, TMT-130, TMT-131, respectively, for the degenerated NP groups. Finally, the mixture was oscillated, centrifuged, incubated for 1 h at RT, and incubated in 5% hydroxylamine (8 μL) for 15 min to terminate the reaction. The samples were stored after lyophilization. 

### 4.6. Liquid Chromatography–Mass Spectrometry/Mass Spectrometry (LC-MS/MS) Analysis

Basic reversed-phase chromatography was conducted on Agilent Zorbax Extend-C18 column (2.1 × 150 mm; 5 μm particle size), using an Agilent 1100 Series HPLC instrument.

The labeled protein samples were loaded at a flow rate of 2 μL/min onto an Acclaim PepMap 100 RP-C18 column (100 μm × 2 cm; Thermo Fisher Scientific, Waltham, MA, USA) and then separated using an Acclaim PepMap RSLC RP-C18 column (75 μm × 15 cm; Thermo Fisher Scientific, Waltham, MA, USA). For the analysis of peptide fractions, mobile phase A was 99.9% water + 0.1% formic acid, and mobile phase B was 80% acetonitrile + 19.9% water + 0.1% formic acid. Full MS scans were acquired in the mass range of 350–1500 *m*/*z* with a mass resolution of 60,000 and the automatic gain control (AGC) target value was set at 3 × 10^6^. The 20 most intense peaks in MS were fragmented with higher-energy collisional dissociation (HCD) with a collision energy of 32. MS/MS spectra were obtained with a resolution of 15,000 with an AGC target of 2 × 10^5^ and a max injection time of 40 ms. The dynamic exclusion was set for 30 s and run under positive mode. 

### 4.7. Database Search and Bioinformatics

The MS/MS data obtained were processed using Proteome Discoverer (v2.4) with specific library search parameters as outlined in Table 2. After obtaining DEPs, gene ontology (GO) annotation and Kyoto Encyclopedia of Genes and Genomes (KEGG) pathway analysis were performed to evaluate their functions. The online analysis software Omicsbean (http://www.omicsbean.cn/, accessed on 25 August 2022) was used for GO/KEGG pathway analysis. Hierarchical clustering analysis was performed using R scripts (ComplexHeatmap 2.4.3). For the GO/KEGG functional enrichment analyses, the specific proteins were used as the background list, and DEPs filtered from the background list were treated as the candidate list. *p*-values were calculated using the hypergeometric distribution test and indicated whether the functional set was significantly enriched in the DEP list. The Benjamini and Hochberg multiplex test was used to adjust the *p*-value and obtain the false discovery rate. 

The STRING database (https://string-db.org/, accessed on 25 August 2022) predicts functional correlations between proteins based on genomic correlations between encoded genes [54,55]. DEPs were analyzed in the STRING database by selecting native or proximate species (blast e-value: 1 × 10^−10^) to obtain their interactions. The Python package ‘network’ was used to visualize the protein interactions.

### 4.8. Extraction of Human-Derived NP Cells 

NP tissues were surgically isolated and minced into 1 mm2 pieces, before being incubated at 37 °C with 5% CO_2_ in DMEM/F-12 complete medium (containing 10% fetal bovine serum) with 0.25 mg/mL type II collagenase for 8 h. The resulting sediment was collected by centrifugation at 800 rpm for 5 min, inoculated in DMEM/F-12 complete medium (containing 10% fetal bovine serum), and cultured further; the medium was changed every 2–3 days. 

### 4.9. Senescence-Associated β-Galactosidase (SA-β-Gal) Staining and Immunofluorescence

SA-β-Gal staining was performed using a cellular senescence β-galactosidase staining kit (Cell Signaling Technology, Boston, MA, USA), and the results were recorded under a microscope (Olympus, Tokyo, Japan).

For immunofluorescence, fixed cells were permeabilized in 0.5% Triton X-100 and blocked with 10% goat serum. Next, the cells were incubated overnight at 4 °C with primary antibodies, including MMP3 (1:50), LUM (1:200), TIMP1 (1:200), and CDC42 (1:200). On the following day, the cells were incubated with secondary antibodies for 1 h at 37 °C. The nuclei were stained with DAPI solution for 10 min, and photographed under a fluorescence microscope (Olympus, Tokyo, Japan).

### 4.10. Statistical Analysis 

Statistical analysis was performed using GraphPad Prism software (version 20.0; GraphPad Software, San Diego, CA, USA). Quantitative data are presented as means ± standard deviations (SDs); Western blotting data were analyzed using a t-test between two groups. *p*-values < 0.05 were considered statistically significant.

## 5. Conclusions

We identified abnormal molecules in aging NP tissues of IDD patients that were significantly associated with the regulation of the extracellular region, collagen-containing ECM, cellular protein metabolic processes, and the phase-mediated IDD signaling pathway. These findings provide a strong foundation for further investigation of the pathogenesis and treatment of IDD in degenerative spinal disease.

## Figures and Tables

**Figure 1 ijms-24-13236-f001:**
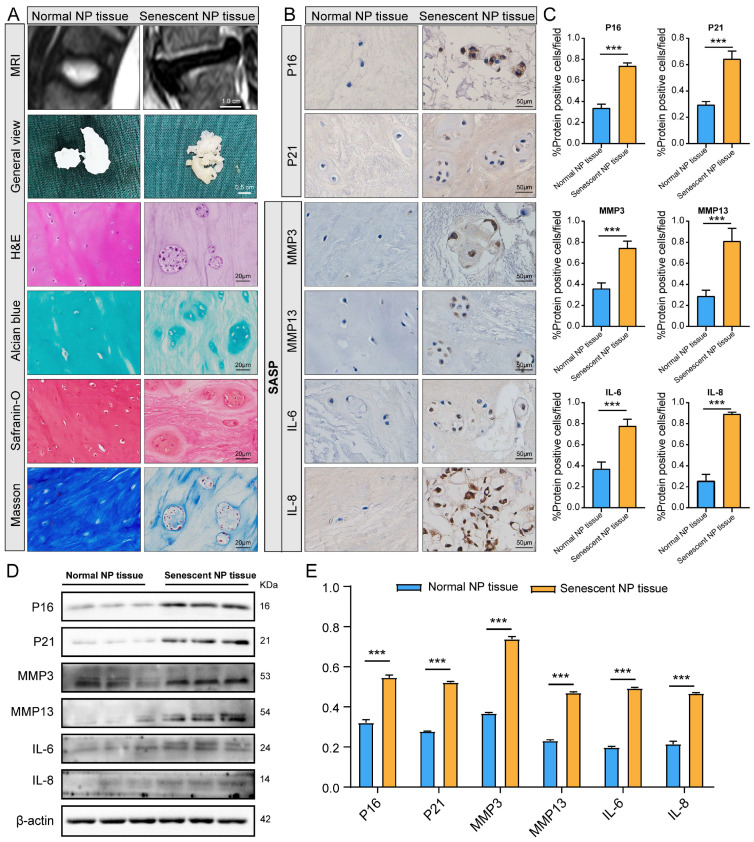
Screening of senescent nucleus pulposus (NP) samples. (**A**) Representative images obtained by magnetic resonance imaging, and staining with hematoxylin and eosin, Alcian blue, Safranin O, and Masson staining of NP tissues obtained from patients with interverbal disc degeneration. (**B**,**C**) Immunohistochemical analysis of P16, P21, and SASP markers (MMP3, MMP13, IL-6, and IL-8) expression in normal and senescent NP tissues. (**D**,**E**) Western blot analysis of P16, P21, and SASP marker (MMP3, MMP13, IL-6, and IL-8) expression in normal and senescent NP tissues. Three biological and three technical replicates were contained in each experiment. *** *p* < 0.001.

**Figure 2 ijms-24-13236-f002:**
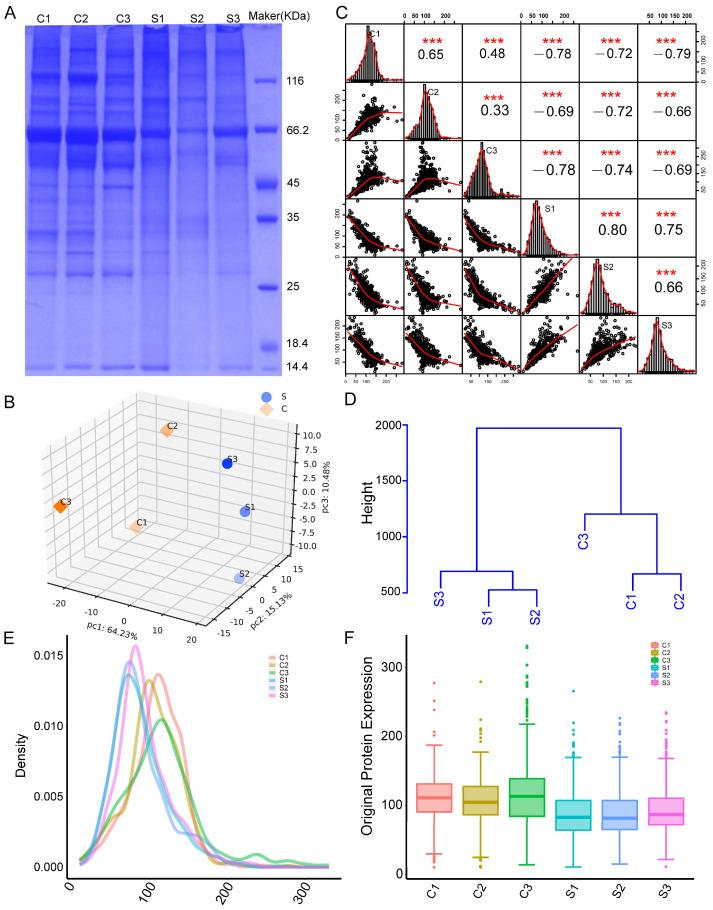
Quality control of nucleus pulposus (NP) samples, credible protein analysis, and other data. (**A**) SDS-PAGE was performed to check the quality of the NP samples. (**B**) 3D scatter plot of the distribution of principal component analysis results of all samples using quantified proteins. (**C**) Pearson’s correlation coefficients were used to examine the pairwise correlations among proteins. (**D**) Hierarchical clustering dendrogram of the Euclidean distances of samples. Each branch terminal represents a sample, and samples clustered in the same branch have similar or close expression characteristics. The Euclidean distance of the ordinate is used to measure the distance between samples not clustered in the same branch. (**E**) Density plot of credible protein expression. The x-axis represents the range of values selected for sample expression, and the y-axis represents the probability density of expression. Each curve represents the probability distribution of the expression value for that sample. Higher “peaks” indicate a greater density of data. (**F**) Box plots of credible protein expression. The height of the box reflects the degree of fluctuation of the data, where flatter boxes indicate more concentrated data. The more the median deviates from the center of the upper and lower quartiles, the stronger the skewness of the distribution. C: control NP tissue; S: senescent NP tissue. *** *p* < 0.001.

**Figure 3 ijms-24-13236-f003:**
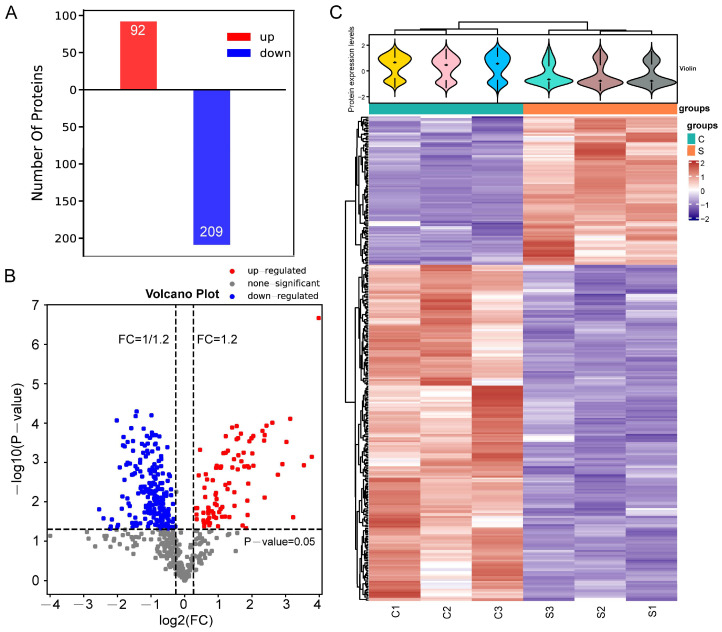
Differentially expressed proteins (DEPs). (**A**) In total, 301 DEPs were identified in aging nucleus pulposus tissues, including 92 upregulated and 209 downregulated proteins. (**B**) Volcano plot of the filtered DEPs. The abscissa of the volcano plots is log2 (fold change), with upregulated proteins on the right and downregulated proteins on the left. The ordinate is -log10 (*p*-value), where higher values indicate greater significance. Blue dots represent downregulated, red dots represent upregulated, and black dots represent nonsignificant DEPs. (**C**) Aggregated heatmap of 301 DEPs. The violin plots in the upper part combine box and density plots. The flatter the violin box, the more centralized the data. The outline of the violin box reflects the probability distribution of the expression values. Different colors represent different samples. The “+” in the middle of the violin plot indicates the median of the data. The vertical axis represents protein expression levels. Below the violin plot are column annotations for the heatmap, with samples in the same group corresponding to the same color block annotation. The aggregation heatmap is clustered according to protein expression level. Red indicates strongly expressed proteins and blue indicates weakly expressed proteins. Each row corresponds to the expression level of a protein in the different groups, and each column indicates the expression levels of all of the DEPs in the different groups.

**Figure 4 ijms-24-13236-f004:**
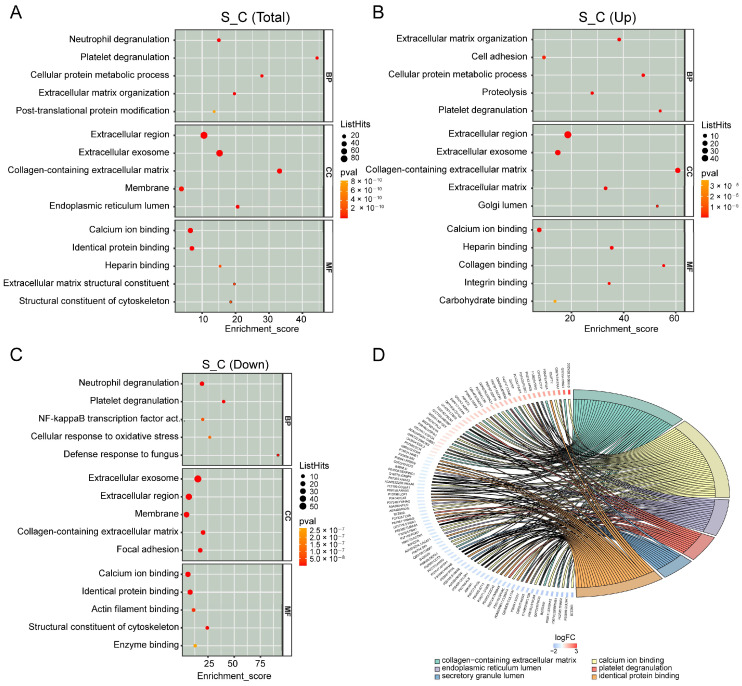
Gene ontology (GO) analysis of differentially expressed proteins (DEPs). (**A**) GO analysis of the top 15 DEPs in terms of biological processes (BP), cellular components (CC), and molecular function (MF). (**B**) GO analysis of the top 15 upregulated DEPs. (**C**) GO analysis of the top 15 downregulated DEPs. (**D**) A chord diagram displaying the GO enrichment analysis of DEPs. The protein is located on the left, with red indicating upregulation and blue indicating downregulation. The selected GO term is shown below the diagram.

**Figure 5 ijms-24-13236-f005:**
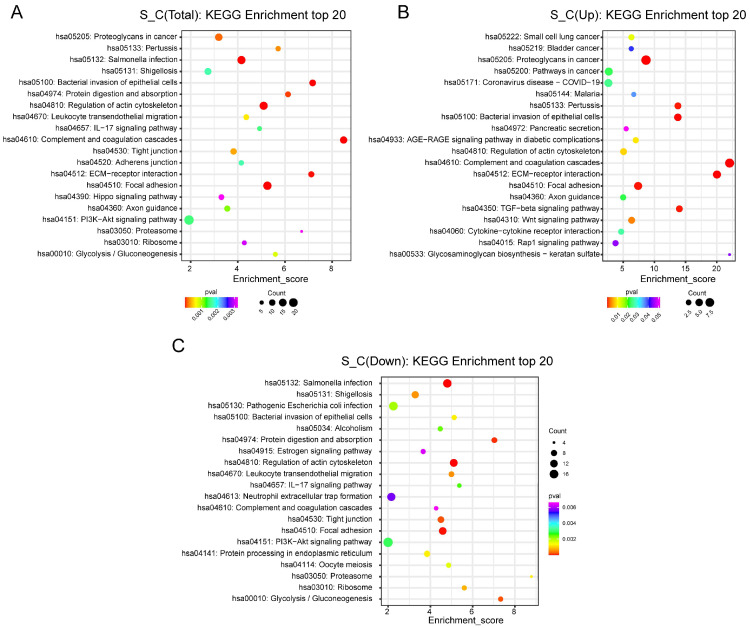
Kyoto Encyclopedia of Genes and Genomes (KEGG) analysis of differentially expressed proteins (DEPs). (**A**) KEGG enrichment of the top 20 DEPs; (**B**) KEGG enrichment of the top 20 upregulated DEPs; (**C**) KEGG enrichment of the top 20 downregulated DEPs. The x-axis represents the enrichment score, and the y-axis represents the pathway information. The bubble size represents the number of differential proteins, with larger bubbles indicating more differential proteins.

**Figure 6 ijms-24-13236-f006:**
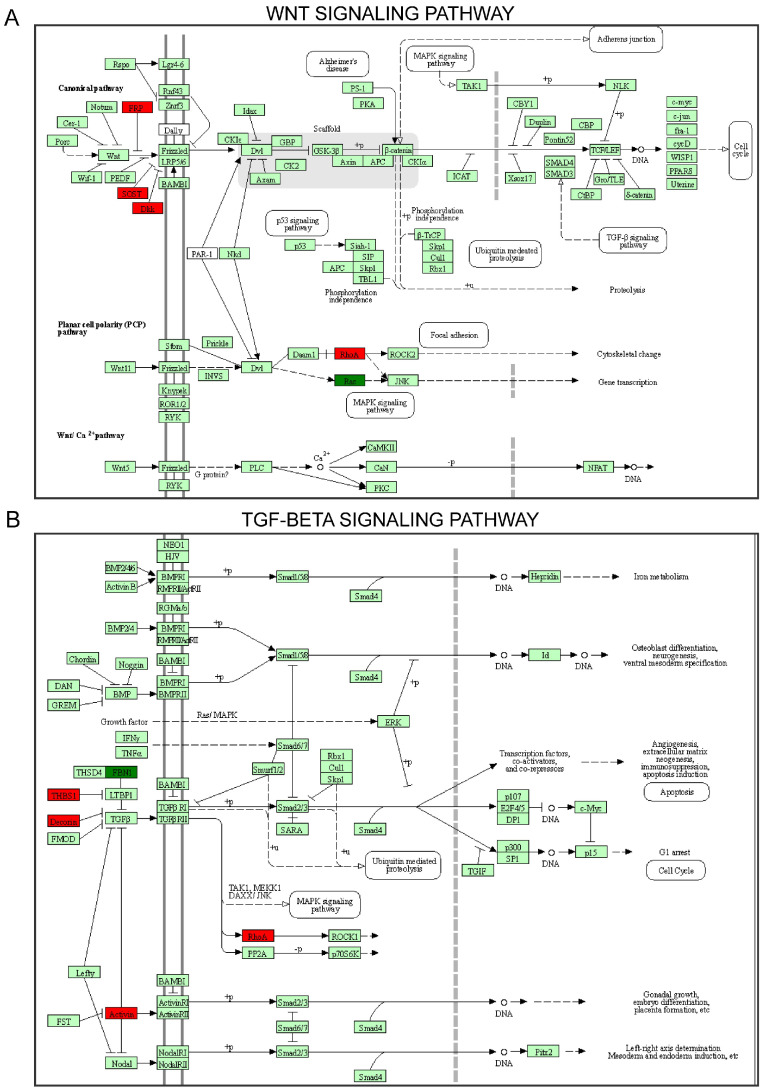
Differentially expressed proteins in the Kyoto Encyclopedia of Genes and Genomes (KEGG) pathway of Wnt and TGF-β signaling pathways. Differentially expressed proteins are represented as boxes and labeled with the gene names. Different colors represent protein expression levels, with red indicating upregulation and green indicating downregulation.

**Figure 7 ijms-24-13236-f007:**
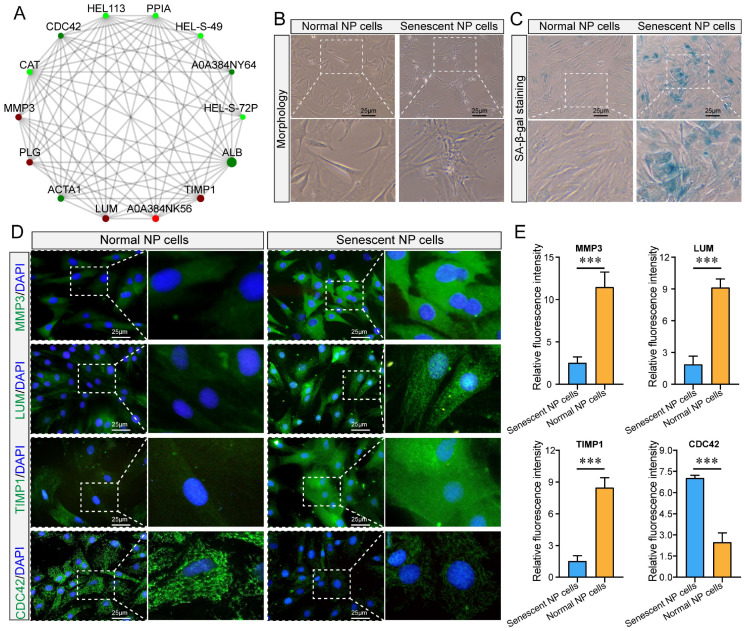
Protein–protein interaction (PPI) network and experimental verification of differentially expressed proteins (DEPs). (**A**) PPI network indicating the interaction between DEPs. Circles represent differential proteins or genes, with red representing upregulated proteins or genes and green representing downregulated proteins or genes. The size of the circles represents the level of connectivity, with larger circles indicating higher connectivity. (**B**,**C**) Gross morphological traits of the nucleus pulposus, SA-β-galactosidase staining, and statistical analysis of cellular senescence levels. (**D**,**E**) Immunofluorescence staining to detect the expression levels of the MMP3, LUM, TIMP1, and CDC42 proteins and statistical analysis. Three biological and three technical replicates were included in each experiment. *** *p* < 0.001.

**Table 1 ijms-24-13236-t001:** Absorbance and concentration of the sample *.

Samples	Absorbance 1	Absorbance 2	Absorbance 3	Average Absorbance	Measured Concentration (μg/μL)	True Concentration (μg/μL)
C1	0.212	0.188	0.204	0.201	0.161	1.607
C2	0.171	0.162	0.182	0.172	0.133	1.326
C3	0.302	0.292	0.280	0.291	0.246	2.460
S1	0.323	0.293	0.285	0.300	0.255	2.545
S2	0.265	0.223	0.260	0.249	0.206	2.062
S3	0.197	0.218	0.170	0.195	0.155	1.547

*: The absorbance values of the test samples were measured, and the average absorbance of OD562 was calculated. The standard curve plot was created with OD562 on the horizontal axis and the standard protein concentration on the vertical axis. According to the regression equation and calculate the concentration of the test sample. Actual concentration = Measured concentration × Dilution ratio. Dilution ratio, 1:10.

**Table 2 ijms-24-13236-t002:** Mass spectrometry retrieval parameters.

Items	Settings
Static modification	TMT (N-term, K); Carbamidomethyl (C)
Dynamic modification	Oxidation (M), Acetyl (N-term)
Digestion	Trypsin
Instrument	Q Exactive HF
MS1 tolerance	10 ppm
MS2 tolerance	0.02 Da
Missed cleavages	2
Database	Uniprot-taxonomy_9606.fasta

## Data Availability

To protect the biological information and privacy of the donors of this study, the raw data are not to be shared publicly but are available on request from the authors.

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
