# Peer review of "TMT-Based Proteomics Analysis of Senescent Nucleus Pulposus from Patients with Intervertebral Disc Degeneration"

_ijms, 2023, doi:10.3390/ijms241713236_

Round 1
Reviewer 1 Report
The authors have used TMT based proteomics to analyze senescent nucleus pulposus from patients with intervertebral disc degeneration. The paper is generally well written and describes a well-designed study into one of the leading causes of lower back pain.
However, I am unable to complete a review with an incomplete description of the method used, and an apparently poor assignment of mass spectra (10,107 out of 472,871).
The manuscript needs to accurately describe the mass spectrometer used, the source, the HPLC system, the exact TMT kit etc. The manuscript needs to describe how individually TMT labeled samples were mixed and how were the final ratios adjusted. The manuscript needs to explain why only 10,107 mass spectra were assigned of a total of 472,871.
Author Response
Response letter
Dear editor and reviewers:
Thank you for your letter and for the reviewers’ comments concerning our manuscript entitled “TMT-based proteomics analysis of senescent nucleus pulposus from patients with intervertebral disc degeneration” (Manuscript ID: ijms-2554560). Those comments are all valuable and very helpful for revising and improving our manuscript, as well as the important guiding significance to our research. We have studied the comments carefully and have made corrections which we hope meet with approval. Revised portions are marked in red in the manuscript. The main corrections in the manuscript and the responses to the reviewers’ comments are the following:
A point-by-point response to the comments of reviewer #1
- The authors have used TMT based proteomics to analyze senescent nucleus pulposus from patients with intervertebral disc degeneration. The paper is generally well written and describes a well-designed study into one of the leading causes of lower back pain. However, I am unable to complete a review with an incomplete description of the method used, and an apparently poor assignment of mass spectra (10,107 out of 472,871). The manuscript needs to accurately describe the mass spectrometer used, the source, the HPLC system, the exact TMT kit etc. The manuscript needs to describe how individually TMT labeled samples were mixed and how were the final ratios adjusted. The manuscript needs to explain why only 10,107 mass spectra were assigned of a total of 472,871.
RE: Thank you for your very meaningful suggestion. We have examined the manuscript carefully, and in the section on methods, we have added relevant contents in detail about the mass spectrometer used, the source, the HPLC system, and the exact TMT kit (lines 345-347).
We have described how individually TMT-labeled samples were mixed and how were the final ratios adjusted (lines 347-352).
As to why only 10,107 mass spectra were assigned of a total of 472,871, you mentioned. Liquid chromatography-mass spectrometry/mass spectrometry (LC-MS/MS) conditions we use are as follows (lines 353-366, 368-369):
The chromatographic conditions: basic reversed-phase chromatography was conducted on Agilent Zorbax Extend-C18 column (2.1×150 mm; 5 μm particle size), using an Agilent 1100 Series HPLC instrument. The labeled protein samples were loaded at a flow rate of 2 μL/min onto an Acclaim PepMap 100 RP-C18 column (100 μm × 2 cm; Thermo Fisher) and then separated using an Acclaim PepMap RSLC RP-C18 column (75 μm × 15 cm; Thermo Fisher). For the analysis of peptide fractions, mobile phase A was 99.9% water + 0.1% formic acid, and mobile phase B was 80% acetonitrile + 19.9% water + 0.1% formic acid.
The mass spectrum conditions:full MS scans were acquired in the mass range of 350 – 1500 m/z with a mass resolution of 60000 and the AGC target value was set at 3e6. The 20 most intense peaks in MS were fragmented with higher-energy collisional dissociation (HCD) with a collision energy of 32. MS/MS spectra were obtained with a resolution of 15000 with an AGC target of 2e5 and a max injection time of 40 ms. The Q Exactive HF dynamic exclusion was set for 30 s and run under positive mode.
Data was analyzed by Proteome Discover 2.4 (Thermo Fisher). The following table shows the specific database search parameters (lines 382-385).
Table 2. Mass spectrometry retrieval parameters.
Items |
Settings |
Static modification |
TMT (N-term, K); Carbamidomethyl(C) |
Dynamic modification |
Oxidation(M),Acetyl(N-term) |
Digestion |
Trypsin |
Instrμment |
Q Exactive HF |
MS1 tolerance |
10ppm |
MS2 tolerance |
0.02Da |
Missed Cleavages |
2 |
Database |
Uniprot-taxonomy_9606.fasta |

Reviewer 2 Report
In the current manuscript, the authors extracted the proteins from tissue samples of IDD patients and quantified them using the TMT-labeling method. The differential protein expression in Healthy and senescent NP tissues was performed by various statistical methods. The authors quantified over 300 proteins across 6 samples. Although, the authors selected three normal and three senescent NP tissue samples, the expressed protein analysis seem ambiguous as the authors did not provide many details. I have listed a few points that could help to improve the manuscript.
1. First of all, the material and Methods section needs to complete rewrite with all the details. Also, the TMT labeling and LC-MS/MS analysis methods are not clear and need more details. Which TMT reagent is used?
2. Is the experiment done in duplicates or triplicates? Biological or technical replication is highly recommended for this type of experiment to draw solid conclusions.
3. Results section- 2.2. NP sample quality control- SDS PAGE image does not show any quality of the proteins, it only shows the presence of proteins. Is there any reference/std protein used to check the quality of the proteins?
4. 2.3. LC-MS/MS analysis- needs thorough rewriting. “In mass spectrometry, we obtained 472,821 secondary spectrograms” does not tell anything.
5. Figure 3 is not necessary. How many proteins are identified and how many are quantified across the samples including replicates is important.
6. Figure 4B Volcano plot DEPs should be Log2 FC more than 1, this will significantly reduce the number of DEPs. The authors draw the line and considered log2 FC 0.1 or 0.2 which are not significant at all. Y-axis should -log10 (p-value). Figure 4C- what is on Y-axis?
7. Section 4.7 Database search and bioinformatics- “The MS/MS data obtained were processed using Proteome Discoverer (v2.4) with specific library search parameters as outlined in Table 1.” I did not find Table 1. The authors could write all the details in this section.
8. Final comment- when the authors use the right log2 FC of 1 or above, all the following results will change.
Need minor language editing.
Author Response
Response letter
Dear editor and reviewers:
Thank you for your letter and for the reviewers’ comments concerning our manuscript entitled “TMT-based proteomics analysis of senescent nucleus pulposus from patients with intervertebral disc degeneration” (Manuscript ID: ijms-2554560). Those comments are all valuable and very helpful for revising and improving our manuscript, as well as the important guiding significance to our research. We have studied the comments carefully and have made corrections which we hope meet with approval. Revised portions are marked in red in the manuscript. The main corrections in the manuscript and the responses to the reviewers’ comments are the following:
A point-by-point response to the comments of reviewer #2
- First of all, the material and Methods section needs to complete rewrite with all the details. Also, the TMT labeling and LC-MS/MS analysis methods are not clear and need more details. Which TMT reagent is used?
RE: Thank you for your valuable suggestion. In the section of material and methods, we have added in detail the relevant contents of the TMT labeling and LC-MS/MS analysis methods (line344-366).
TMT labeling was performed for mass spectroscopy (MS) using a TMTsixplex™ Label Reagent Set according to the manufacturer’s protocol for the 6-plex TMT kit (Thermo Fisher Scientific, USA) (lines 344-346).
- Is the experiment done in duplicates or triplicates? Biological or technical replication is highly recommended for this type of experiment to draw solid conclusions.
RE: Thank you for your very meaningful suggestions for this section. As you mentioned, biological or technical replication is important for this type of experiment to draw solid conclusions. Three biological and three technical replicates were contained in each experiment. We have explained this in the relevant parts of the manuscript (lines 95, and 222-223).
- Results section- 2.2. NP sample quality control- SDS PAGE image does not show any quality of the proteins, it only shows the presence of proteins. Is there any reference/std protein used to check the quality of the proteins?
RE: Thank you for your very meaningful suggestions for this section. To start, we measure the absorbance values of the test samples and calculate their average OD562. Next, we create a standard curve plot with OD562 on the horizontal axis and the standard protein concentration on the vertical axis. We obtain the regression equation, which is the standard curve as shown in the figure below. Lastly, we calculate the concentration of the test sample by multiplying the measured concentration with the dilution factor to obtain the actual concentration of the sample (lines 97-101, and 107-108).
Table 2. Absorbance and concentration of the sample
Samples |
Absorbance 1 |
Absorbance 2 |
Absorbance 3 |
Average absorbance |
measured concentration(ug/uL) |
True concentration(ug/uL) |
C1 |
0.212 |
0.188 |
0.204 |
0.201 |
0.161 |
1.607 |
C2 |
0.171 |
0.162 |
0.182 |
0.172 |
0.133 |
1.326 |
C3 |
0.302 |
0.292 |
0.280 |
0.291 |
0.246 |
2.460 |
S1 |
0.323 |
0.293 |
0.285 |
0.300 |
0.255 |
2.545 |
S2 |
0.265 |
0.223 |
0.260 |
0.249 |
0.206 |
2.062 |
S3 |
0.197 |
0.218 |
0.170 |
0.195 |
0.155 |
1.547 |
To further evaluate protein quality control, SDS-PAGE electrophoresis was applied. The results showed that the protein did not degrade since the protein bands were clear and uniform, which could meet the experimental needs.
- 2.3. LC-MS/MS analysis- needs thorough rewriting. “In mass spectrometry, we obtained 472,821 secondary spectrograms” does not tell anything.
RE: Thank you for your careful review and nice suggestion. After re-reading and checking this section, we found that our description is not clear. We have carefully re-examined the content of this section and rewrote it to a more perfect one (lines 109-113)
- Figure 3 is not necessary. How many proteins are identified and how many are quantified across the samples including replicates is important.
RE: Thank you for your valuable suggestion. As you mentioned, how many proteins are identified and how many are quantified across the samples including replicates is important. Figure 3 showed the distribution of peptide numbers corresponding to each qualitative protein and the number of proteins corresponding to the molecular weights. During the qualitative process, we compared each peptide with a peptide in the background database; using the database search software, we obtained the coverage index of this peptide relative to the complete protein sequence. We analyzed the coverage index and rate of peptide coverage.
- Figure 4B Volcano plot DEPs should be Log2 FC more than 1, this will significantly reduce the number of DEPs. The authors draw the line and considered log2 FC 0.1 or 0.2 which are not significant at all. Y-axis should -log10 (p-value). Figure 4C- what is on Y-axis?
RE: Thank you for pointing out the mistakes which we have corrected. After re-reading and checking Figure 4, We have changed "log10 (p-value)" into "-log10 (p-value)". In Figure 4C, the violin plots in the upper half combine box and density plots. The flatter the violin box, the more centralized the data. The outline of the violin box reflects the probability distribution of the expression values. Different colors represent different samples. The “+” in the middle of the violin plot indicates the median of the data. The vertical axis represents protein expression levels (Figure 4C, lines 165-168).
- Section 4.7 Database search and bioinformatics- “The MS/MS data obtained were processed using Proteome Discoverer (v2.4) with specific library search parameters as outlined in Table 1.” I did not find Table 1. The authors could write all the details in this section.
RE: Thank you for pointing out the mistakes which we have corrected. We have added Table 2 to the manuscript (lines 382-385).
- Final comment- when the authors use the right log2 FC of 1 or above, all the following results will change.
RE: Thank you again for your hard work on this manuscript. DEPs between the groups were screened using fold-change≥1.2 and p < 0.05. In future studies, we will select genes with significant difference multiples and play a key role in the progression of IDD.
- Comments on the Quality of English Language: Need minor language editing.
RE: Thank you for your kind suggestion. We carefully read and re-examined the language of the manuscript. We have made corrections to the organization and spelling of the language. We thank Textcheck (www.textcheck.com) for English language editing. The certificate materials are as follows.

Round 2
Reviewer 1 Report
I have no further criticisms.
Author Response
Thank you very much for your approval of this manuscript.
Reviewer 2 Report
The revised version seems improved; however, the authors did not touch the main results derived from the wrong log2 FC. I have listed a few points that could help to improve the manuscript.
1. The newly added TMT and LC-MS method is enough but needs more details such as which sample was labeled with a specific label.
2. Related to my second comment- Table 1, measured concentration and true concentration are confusing. Mention only one or explain what is true concentration and how it was calculated in the table footnote and not in the results.
3. My third comment- 2.3. LC-MS/MS analysis- needs thorough rewriting. The whole section really needs rewriting. The current statements do not make any sense.
4. No satisfactory explanation from the author on this comment- Figure 3 is not necessary. How many proteins are identified and how many are quantified across the samples including replicates is important.
5. No satisfactory explanation from the author on this comment- Figure 4B Volcano plot DEPs should be Log2 FC more than 1, this will significantly reduce the number of DEPs. The authors draw the line and considered log2 FC 0.1 or 0.2 which are not significant at all.
6. Final comment- when the authors use the right log2 FC of 1 or above, all the following results will change. The authors did not change the log2FC.
minor grammar and spelling check.
Author Response
Response letter
Dear editor and reviewers:
Thank you for your letter and for the reviewers’ comments concerning our manuscript entitled “TMT-based proteomics analysis of senescent nucleus pulposus from patients with intervertebral disc degeneration” (Manuscript ID: ijms-2554560). Those comments are all valuable and very helpful for revising and improving our manuscript, as well as the important guiding significance to our research. We have studied the comments carefully and have made corrections which we hope meet with approval. Revised portions are marked in red in the manuscript. The main corrections in the manuscript and the responses to the reviewers’ comments are the following:
A point-by-point response to the comments of reviewer #2
- The newly added TMT and LC-MS method is enough but needs more details such as which sample was labeled with a specific label.
RE: Thank you for your comment. We carefully read and re-examined the details of the TMT and LC-MS method. The peptides of the digested NP samples were labeled with TMT-126, TMT-127, and TMT-128 for the normal NP groups, respectively; TMT-129, TMT-130, TMT-131 for the degenerated NP groups, respectively. In the revised manuscript, we have added more details in TMT and LC-MS method (lines 342-344).
- Related to my second comment- Table 1, measured concentration and true concentration are confusing. Mention only one or explain what is true concentration and how it was calculated in the table footnote and not in the results.
RE: Thank you for your valuable suggestion. We have explained what is true concentration and how it was calculated in the table footnote (lines 102-103).
- My third comment- 2.3. LC-MS/MS analysis- needs thorough rewriting. The whole section really needs rewriting. The current statements do not make any sense.
RE: Thank you for your very meaningful suggestions for this section. We have redescribed the results of LC-MS/MS analysis and deleted the meaningless content (lines 103-106).
- No satisfactory explanation from the author on this comment- Figure 3 is not necessary. How many proteins are identified and how many are quantified across the samples including replicates is important.
RE: Thank you again for your kind suggestion. As you mentioned, figure 3 is not necessary. We have used figure 3 as supplementary material.
- No satisfactory explanation from the author on this comment- Figure 4B Volcano plot DEPs should be Log2 FC more than 1, this will significantly reduce the number of DEPs. The authors draw the line and considered log2 FC 0.1 or 0.2 which are not significant at all.
RE: Thank you for pointing this out. logFC, FC (Fold Change) is the change multiple of DEPs, indicating the differential expression multiple of the experimental group compared with the control group, and then log. Generally, the value of FC is greater than 2 times (log2 FC=1) is meaningful, and it is acceptable to relax the requirement of 1.5 times (log2 FC=0.585) or 1.2 times (log2 FC=0.263). In this study, consistent with most literature reports [1-5], considering that some proteins were not significantly up-regulated or down-regulated, but played a crucial role in organisms, we selected the fold change >1.2(logFC=0.263) or <1/1.2 (logFC=-0.263) and p < 0.05 were considered as DEPs for further analysis.
Reference:
[1] Yang L, Hou A, Zhang X, Zhang J, Wang S, Dong J, Zhang S, Jiang H, Kuang H. TMT-based proteomics analysis to screen potential biomarkers of Achyranthis Bidentatae Radix for osteoporosis in rats. Biomed Chromatogr. 2022 Apr;36(4):e5339. doi: 10.1002/bmc.5339. Epub 2022 Feb 2. PMID: 35043449.
[2] Ye J, Li Y, Kong C, Ren Y, Lu H. Label-free proteomic analysis and functional analysis in patients with intrauterine adhesion. J Proteomics. 2023 Apr 15;277:104854. doi: 10.1016/j.jprot.2023.104854. Epub 2023 Feb 24. PMID: 36841354.
[3] Cheng L, Wang X, Chou H, Liu T, Fu H, Li G. Proteomic Sequencing of Stellate Ganglions in Rabbits With Myocardial Infarction. Front Physiol. 2021 Dec 16;12:687424. doi: 10.3389/fphys.2021.687424. PMID: 34975513; PMCID: PMC8716754.
[4] Luo W, Yang Z, Zhang W, Zhou D, Guo X, Wang S, He F, Wang Y. Quantitative Proteomics Reveals the Dynamic Pathophysiology Across Different Stages in a Rat Model of Severe Traumatic Brain Injury. Front Mol Neurosci. 2022 Jan 25;14:785938. doi: 10.3389/fnmol.2021.785938. PMID: 35145378; PMCID: PMC8821658.
[5] Yu XT, Wang F, Ding JT, Cai B, Xing JJ, Guo GH, Guo F. Tandem mass tag-based serum proteomic profiling revealed diabetic foot ulcer pathogenesis and potential therapeutic targets. Bioengineered. 2022 Feb;13(2):3171-3182. doi: 10.1080/21655979.2022.2027173. PMID: 35068329; PMCID: PMC8974021.
- Final comment- when the authors use the right log2 FC of 1 or above, all the following results will change. The authors did not change the log2FC.
RE: Thank you again for your hard work on this manuscript. In this study, consistent with most literature reports [1-5], considering that some proteins were not significantly up-regulated or down-regulated, but played a crucial role in organisms, we selected the fold change >1.2(logFC=0.263) or <1/1.2 (logFC=-0.263) and p < 0.05 were considered as DEPs for further analysis. If you strongly suggest log2 FC of 1 or above in this manuscript, we will re-analyze the data from this study.
Reference:
[1] Yang L, Hou A, Zhang X, Zhang J, Wang S, Dong J, Zhang S, Jiang H, Kuang H. TMT-based proteomics analysis to screen potential biomarkers of Achyranthis Bidentatae Radix for osteoporosis in rats. Biomed Chromatogr. 2022 Apr;36(4):e5339. doi: 10.1002/bmc.5339. Epub 2022 Feb 2. PMID: 35043449.
[2] Ye J, Li Y, Kong C, Ren Y, Lu H. Label-free proteomic analysis and functional analysis in patients with intrauterine adhesion. J Proteomics. 2023 Apr 15;277:104854. doi: 10.1016/j.jprot.2023.104854. Epub 2023 Feb 24. PMID: 36841354.
[3] Cheng L, Wang X, Chou H, Liu T, Fu H, Li G. Proteomic Sequencing of Stellate Ganglions in Rabbits With Myocardial Infarction. Front Physiol. 2021 Dec 16;12:687424. doi: 10.3389/fphys.2021.687424. PMID: 34975513; PMCID: PMC8716754.
[4] Luo W, Yang Z, Zhang W, Zhou D, Guo X, Wang S, He F, Wang Y. Quantitative Proteomics Reveals the Dynamic Pathophysiology Across Different Stages in a Rat Model of Severe Traumatic Brain Injury. Front Mol Neurosci. 2022 Jan 25;14:785938. doi: 10.3389/fnmol.2021.785938. PMID: 35145378; PMCID: PMC8821658.
[5] Yu XT, Wang F, Ding JT, Cai B, Xing JJ, Guo GH, Guo F. Tandem mass tag-based serum proteomic profiling revealed diabetic foot ulcer pathogenesis and potential therapeutic targets. Bioengineered. 2022 Feb;13(2):3171-3182. doi: 10.1080/21655979.2022.2027173. PMID: 35068329; PMCID: PMC8974021.
Thank you again for your hard work on this manuscript. We tried our best to improve the manuscript and made some changes in the manuscript. And revised portions are marked in red in the manuscript. We appreciate for editors'/reviewers’ warm work earnestly, and hope that the correction will meet with approval. Once again, thank you very much for your comments and suggestions.
Yours sincerely,
Xueweng Kang
Department of Orthopaedics, Lanzhou University Second Hospital
No. 82 of Linxia Street, Chengguan District, Lanzhou 730030, Gansu, China.
86-13919026469
ery_kangxw@lzu.edu.cn

Round 3
Reviewer 2 Report
The revised version looks much improved and can be accepted.